# Changes in Microbiota Composition during the Anaerobic Digestion of Macroalgae in a Three-Stage Bioreactor

**DOI:** 10.3390/microorganisms12010109

**Published:** 2024-01-05

**Authors:** Dovilė Vasiliauskienė, Mantas Pranskevičius, Regimantas Dauknys, Jaunius Urbonavičius, Juliana Lukša, Vadym Burko, Alvydas Zagorskis

**Affiliations:** 1Department of Chemistry and Bioengineering, Vilnius Gediminas Technical University, Sauletekio av. 11, 10223 Vilnius, Lithuania; dovile.vasiliauskiene@vilniustech.lt (D.V.); jaunius.urbonavicius@vilniustech.lt (J.U.); juliana.luksa@vilniustech.lt (J.L.); 2Institute of Environmental Protection, Vilnius Gediminas Technical University, 10223 Vilnius, Lithuania; mantas.pranskevicius@vilniustech.lt; 3Department of Environmental Protection and Water Engineering, Vilnius Gediminas Technical University, 10223 Vilnius, Lithuania; regimantas.dauknys@vilniustech.lt; 4Laboratory of Genetics, Nature Research Centre, 08412 Vilnius, Lithuania; 5Department of Primary Science Institute of Modern Technologies, Pryazovskyi State Technical University, 87555 Mariupol, Ukraine; burko@mariupol.org.ua

**Keywords:** biogas, methane, macroalgae, three-stage bioreactor, microbiota

## Abstract

The use of microalgae as a raw material for biogas production is promising. Macroalgae were mixed with cattle manure, wheat straw, and an inoculant from sewage sludge. Mixing macroalgae with co-substrates increased biogas and methane yield. The research was carried out using a three-stage bioreactor. During biogas production, the dynamics of the composition of the microbiota in the anaerobic chamber of the bioreactor was evaluated. The microbiota composition at different organic load rates (OLRs) of the bioreactor was evaluated. This study also demonstrated that in a three-stage bioreactor, a higher yield of methane in biogas was obtained compared to a single-stage bioreactor. It was found that the most active functional pathway of methane biosynthesis is PWY-6969, which proceeds via the TCA cycle V (2-oxoglutarate synthase). Microbiota composition and methane yield depended on added volatile solids (VS_added_). During the research, it was found that after reducing the ORL from 2.44 to 1.09 kg VS/d, the methane yield increased from 175.2 L CH_4_/kg VS_added_ to 323.5 L CH_4_/kg VS_added_.

## 1. Introduction

The production of biogas using agricultural and eutrophication products such as the algae *Saccorhiza polyschides* allows the simultaneous digestion of organic waste and the generation of a multipurpose energy carrier (methane) that is further converted into electricity and heat [1,2,3]. Biogas is produced during the anaerobic digestion of organic matter. Anaerobic digestion consists of four stages: hydrolysis, acidification, acetate production, and methanogenesis. All steps are performed by highly specialized and complex microbial communities, and the roles of each member in a consortium are different [4]. These complex interspecies relationships hamper the investigation of microbial communities using traditional microbiological methods [5]. To solve this problem, it is necessary to go beyond the simple identification of microbial species in the methanogenic microbiota to reveal their functional roles during biogas production. The use of macroalgal biomass for biogas production also has its challenges. Salinity can be a serious barrier to microbiota activity. Also, seasonal dispersion may affect the growth and quality of macroalgal biomass. The use of macroalgae found in freshwater bodies or low-salinity seas, such as the Sea of Azov, for biogas production could reduce this risk. It is also necessary to ensure proper biomass collection.

Anaerobic digestion of a substrate for biogas production is affected by many factors. One key factor is organic load rate (OLR). OLR depends on the physicochemical parameters of the substrate, such as the C:N ratio, total solids (TS), volatile solids (VS), and hydraulic retention time (HRT) [6]. Therefore, the optimal OLR allows for controlling biogas production [7]. There are many studies showing that macroalgae can be used for biogas production in biopower plants due to their nutritional properties. Due to the high carbon-nitrogen ratio (C:N), they should be mixed with materials with a low C:N ratio, such as cattle manure, to achieve the optimal C:N ratio. The optimal C:N for biogas production should be 20:1 to 30:1 [8,9].

Biogas produced using macroalgae with co-substrates can be a good alternative to fossil fuels. Macroalgae contain high levels of cellulose, which would affect the yield of biogas during its production. According to previous experiments performed, the most promising approach to pretreatment, at least for brown algae, is a hydrothermal one. Using this method, high amounts of methane were produced, and a positive energy balance was obtained [10]. Another type of experiment revealed a high yield of methane in biogas produced using macroalgae, *Spirogyra varians*. The untreated methane yield was 64.7%, and the pretreatment even increased the methane yield percentage to 69.4%. Due to their rapid biomass production and their suitable composition, these macroalgae are widely used for biogas production [11]. Organic matter is mineralized during anaerobic digestion. They are mineralized in the absence of inorganic oxidants such as nitrate, sulfate, iron, etc. Over 67% of acetaclastic and up to 33% of hydrogenotrophic methanogens produce methane when cellulose degrades [12]. Methanogenic archaea reduce to methane formate, CO_2_, acetate, methylamines, methanol, methyl sulfides, and C1 and C2 compounds. Methanogenic archaea obtain energy during the reduction process. Methanogens also use ingredients such as secondary alcohols and methoxylated aromatic substances (in methoxydotrophic methanogenesis). The methanogenic lineages of Euryarchaea are broadly characterized as hydrogenotrophic (using H_2_ and CO_2_), acetalastic (acetate), methylotrophic (X-CH_3_), and methylotrophic (using H_2_ and X-CH_3_) lineages. The orders Methanococcales, Methanobacteriales, Methanocellales, Methanomicrobiales, and Methanopyrales consist of strict hydrogenotrophic methanogens [13]. Numerous microbial communities regulate the stages of anaerobic digestion: hydrolysis, acidogenesis, acetogenesis, and matenogenesis. They work in a symbiotic relationship. These communities change as parameters change. They depend on the substrates used for anaerobic digestion.

For biogas production using macroalgae and co-substrates, single- or two-stage bioreactors are usually used [14,15]. A two-stage bioreactor can produce about 30% more energy than a single-stage bioreactor [16]. High methane yields can be achieved using a three-stage bioreactor. To obtain good methane production, the anaerobic conditions should be stable, and oxygen levels should be as low as possible. Thus, the intermediate chambers of three-stage bioreactors increase the dissolved oxygen reduction efficiency of the substrate by 80.5% [17]. In the first chamber of the bioreactor, the primary crushing and mixing of biomass are carried out. The prepared substrate then enters the second chamber along with excess oxygen. In the second chamber, the biomass is preheated, and one-third of the biomass always remains, ensuring that oxygen does not enter the third chamber and that anaerobic conditions are maintained in it. The third chamber is used for the digestion process and biogas production.

The aim of this study is to determine and evaluate the composition of the microbiota in a three-stage bioreactor when a substrate, consisting of a mixture of macroalgae, cattle manure with wheat straw, and sewage sludge, is anaerobically treated at different OLRs. Currently, there is insufficient information on changes in microbiota in a three-stage bioreactor when a substrate, consisting of a mixture of macroalgae, cattle manure, and sewage sludge, is anaerobically treated at different organic loadings. The present research would allow a better understanding of the microbiological processes taking place in a three-stage bioreactor, where macroalgae participate in anaerobic digestion.

## 2. Materials and Methods

### 2.1. Substrate and Inoculum

The macroalgae *Cladophora glomerata* (MA) grows abundantly in both marine and freshwater. For this study, macroalgae were taken from the surface waters of the freshwater Šventoji River in Lithuania. These macroalgae were mixed with cattle manure and wheat straw (CM) for testing. Sewage sludge (WS) was used as an inoculant. WS was used as a source of activated methanogenic archaea. These species were used for the first time to determine the biogas yield at different organic loading rates (OLR).

Cattle manure with wheat straw was obtained from a cattle farm, and activated sewage sludge was obtained from anaerobic treatment plants for municipal sewage sludge. Total particulate solids (TS), inorganic solids (NS), and volatile solids (VS) in each feedstock were determined prior to substrate preparation for anaerobic digestion. The TS, VS, and NS of the raw materials were determined using standardized methods [2]. Samples of 5 g were taken for the determination of raw material TS. They were dried at 105 °C for 12 h. The samples were placed in a blast furnace after drying. The samples were heated for 1 h at a temperature of 550 °C in a foaming oven. The VS and NS of raw materials are determined by the weight difference method. The loss mass is equal to VS, and the remaining mass is equal to NS [18,19].

Physical (TS, NS, and VS) and chemical (C, N, H, S, and pH) parameters of the substrate were determined before the test and after each OLR phase. TS, VS, and NS were determined by the standardized methods discussed above. The elemental composition (C, N, H, and S) of the substrate was determined using an EA 3000 elemental composition analyzer (Eurovector, Pavia, Italy). Substrate pH was determined using a Multi-Seven pH meter (Mettler Toledo Solutions, Greifensee, Switzerland). Physicochemical parameters of macroalgae, co-substrate, and cattle manure, along with wheat straw and inoculum from sewage sludge, are presented in Table 1.

### 2.2. Measurement of the Protein, Total Lipids, and Glucose Concentration

Protein, sugars (released glucose), and total lipids content were determined before and after different OLR phases. Substrate analysis was performed in different phases, focusing on the upper (U) and lower (L) layers of the third anaerobic chamber. In order to effectively assess the distribution of solid particles and the studied parameters in the volume of the third anaerobic chamber, samples were taken at different heights (layers) of the chamber. Despite continuous mixing, a possible higher sedimentation of particles in the bottom layer was estimated in this way. Substrate samples for research were collected at three main time points: before methanogenesis from different layers (1U and 1L), when OLR phase was 2.44 kg VS/d (2U and 2L), and when OLR phase was 1.09 kg VS/d (3U and 3L).

Protein content was determined using the Bradford method in triplicate. Substrate samples for research were taken before methanogenesis and after different OLRs. Substrate biomass (50 mg) was mixed with 4 mL of distilled water. The mixture was then inoculated for 24–48 h at 4 °C. The mixture was ultrasonicated in water three times for 30 min at 4 °C. After sonication, the mixture was centrifuged for 10 min (at 4 °C, 10,000× *g*). Protein content was then determined using a BioTek Eon microplate spectrophotometer (BioTek Instruments, Winooski, VT, USA) and measuring absorbance at 595 nm.

The Folch method [20] was used to determine total lipids. Biomass was dried in a drying oven for the research. For this, 50 mL of solvent consisting of chloroform and methanol (volume ratio 2:1) was added to 1 g of dried biomass. The substrate was then destroyed by sonication in water three times for 30 min at room temperature. The solvent with the extracted lipids was vacuum-filtered. The solvent was then evaporated in a drying oven at 60 °C. In total, 20 mL of solvent was evaporated. Total lipid content was determined gravimetrically and expressed as a percentage of dry weight using Equation (1):(1)$$x=\frac{\left((m_{1}-m_{0})\cdot 50\cdot 100\right)}{20\cdot m},\%$$
where *m*_0_ is the empty plate weight, *m*_1_ is the plate weight with the lipids, *m* is the lipid weight, 20 is milliliters of the organic solvent evaporated, and 50 is milliliters of the total organic solvent.

Sugars (released glucose) were measured in triplicate using 3,5-DNS acid (Alfa Aesar, Ward Hill, MA, USA) at 540 nm with the same microplate spectrophotometer. Extraction was performed using the following measures: 50 mg of substrate was placed in a 100 mL flask and mixed with 50 mL of a 2% H_2_SO_4_ solution at a pH of 1.4. This mix was autoclaved at 121 °C for 15 min.

### 2.3. Experimental Setup and Procedure

Macroalgae were collected in containers for testing. They were stabilized by drying and freezing before being fed into the bioreactor. Macroalgae taken from the refrigerator were mixed with co-substrate and inoculum. The biomass was crushed to 5 mm particles using a chopper before being fed into the bioreactor. A three-stage bioreactor was used for the research (Figure 1).

The design of the bioreactor allows for optimization of the biogas production process. The biogas production process using a bioreactor consists of biomass preparation (stage I), optimization of biomass parameters (stage II), and biogas production (stage III). One motor rotates the stirring blades in all bioreactor chambers. Therefore, energy consumption for mixing is saved. Mixer blades installed in all chambers stirred the substrate around the clock in continuous mode at a constant rotation speed of 5.0 rpm.

There are three stages in the bioreactor: the preparation of the substrate stage, a transition stage where the substrate changes from an aerobic (saturated with dissolved oxygen) state to an anaerobic (no oxygen) state, and a methanogenesis stage. The three-stage bioreactor is composed of three chambers: the preparation chamber (60 L capacity) for mixing the substrate, the secondary aerobic-anaerobic chamber (60 L capacity), and the third anaerobic chamber (300 L capacity). The primary mixing of the biomass took place in the preparation chamber. Each time biomass is loaded into the chamber, excess oxygen enters with the substrate. The preparation chamber prevents oxygen from entering the second chamber together with the biomass. From the preparation chamber, the biomass is fed to the secondary chamber, where the substrate is shifted from aerobic (containing dissolved oxygen) to anaerobic conditions [21]. Previous research has shown that methanogenesis works effectively when the temperature is maintained at 37 °C [4,22,23]. The mixer mixes secondary and tertiary chamber substrates simultaneously. From the tertiary chamber, the substrate is removed through the outlet. The biogas is pumped from the secondary and tertiary chambers into the biogas storage tank for composition analysis.

The substrate was initially mixed and homogeneous in a primary preparation chamber before being supplied to the secondary chamber, where it was heated to 37 ± 0.2 °C. From this secondary chamber, where there is a decreased level of oxygen, the substrate was supplied via autophagy to the tertiary anaerobic chamber. The composition of the substrate was: 80.91 ± 0.32 g TS/L and 40.42 ± 0.33 g VS/L. The C:N ratio of the substrate was 26, and the pH value reached 7.77 ± 0.2.

Since the volume of the primary and secondary is 60 L each, the filling of the 300 L with substrate is completed in five 60 L increments. After the loading of the bioreactors, the process was carried out in batch mode. After loading the substrate into the tertiary chamber, no further supply was made until biogas production was stabilized. The substrate was fed to the bioreactor and remained there for 20 days. Constant temperature and mixing modes were ensured in the bioreactor during the adaptation period. Biogas yield and composition were determined. After stabilization of the process on days 21–34, the OLR in the bioreactor was 2.44 kg VS/d. On days 35–50, the OLR in the bioreactor was 1.09 kg VS/d. Samples for microbiota studies were taken before digestion and after stabilization of methane yield at different OLRs on days 34 and 50, respectively, in the third anaerobic chamber. OLR was guaranteed when the supply capacity (SC) of the substrate reached 60 L/d and 30 L/d, respectively. The total volume of the substrate during all phases was 300 L. The three-stage bioreactor operational parameters for the AD process are presented in Table 2.

The production and quality of biogas were determined each day. For the evaluation of biogas composition, concentrations of CH_4_, CO_2_, O_2_, and H_2_S in biogas were determined using the GFM 410 analyzer (Gas Data Limited, Coventry, UK). The biogas composition was periodically checked using a gas chromatograph HP 5890 Series II single injector with a single FID detector (Hewlett Packard, Palo Alto, CA, USA).

### 2.4. DNA Isolation, Sequencing, and Bioinformatics Analysis

The genomic DNA of the microbial community samples was extracted and purified before methane synthesis (1U and 1L) and at the different OLRs (2.44 and 1.09 kg VS/d) using the PureLink Microbiome DNA Purification Kit (Invitrogen, Waltham, MA, USA), according to the manufacturer’s instructions. Samples for metagenomic analysis were collected in a third anaerobic chamber.

Genomic DNA samples were profiled using shotgun metagenomic sequencing. Samples were profiled to generate a metagenomic library. Two different approaches were used to construct a shotgun metagenomic library. The first method involved the use of the KAPA™ HyperPlus Library Preparation Kit (KAPA Biosystems, Wilmington, MA, USA) with an input of up to 100 ng of DNA. Internal single-index 8 bp barcodes in combination with TruSeg^®^ adapters (Illumina, San Diego, CA, USA) were used in this approach. The second method involved using the Nextera^®^ DNA Flex Library Prep Kit (Illumina, San Diego, CA, USA) with an input of up to 100 ng of DNA. This method used internal dual-index 8 bp barcodes with Nextera^®^ adapters (Illumina, San Diego, CA, USA). All prepared libraries were quantified using TapeStation^®^ (Agilent Technologies, Santa Clara, CA, USA). Libraries were pooled in equal parts after quantification. The resulting library set was quantified using qPCR based on the Zymo Research protocol. The prepared library was sequenced using the Illumina HiSeq^®^ or Illumina NovaSeg^®^ platforms at Zymo Research (Irvine, CA, USA).

Raw sequencing reads were trimmed for bioinformatics analysis. Low-quality sequences and adapters were removed using Trimmomatric-0.33 [24]. A sliding window (6 bp size and 20 bp quality limits) was used for quality trimming. Reads shorter than 70 bp were removed. The sequence aligner DIAMOND was used to identify antimicrobial resistance and virulence factor genes [25]. Microbial composition was profiled using MetaPhlAn4 [26]. Once the taxonomy and abundance information were obtained, they were analyzed. Alpha and beta diversity analysis was performed; microbial taxonomic assignment in QIIME [27], taxon abundance heatmap with hierarchical clustering [28], and biomarker discovery using LEfSe [29] with default settings (*p* > 0.05 and LDA effect size > 2) were performed. Humman2 was used for functional profiling, identification of the UniRef gene family, and MetaCyc metabolic pathways [30].

## 3. Results and Discussion

### 3.1. Changes in Substrate Composition during Biogas Production

The concentrations of proteins, lipids, and polysaccharides play an important role during the methanogenic process. We determined their concentrations in the substrate before incubation in the bioreactor and after stabilization of the process at 5 days of retention (when the OLR was 2.44 kg VS/d) and at 10 days of retention (when the OLR was 1.09 kg VS/d) (Table 3). Before methanogenesis, the total lipid content of the substrate was recorded at 1.49%. Subsequently, during the retention period of 5 days, when the OLR was 2.44 kg VS/d, there was a notable reduction of 1.3% in the total lipid content, which further decreased by 0.4% after 10 days, when the OLR was 1.09 kg VS/d. It is noteworthy that the initial substrate exhibited the highest total lipid content. This observation raises the possibility that the initial homogenization process in the first bioreactor stage may have been less effective. A possible reason may be the substrate fraction size, which ranged from 0 to 1 mm, hindering the even distribution of lipids. Therefore, it is imperative to improve the homogenization process, as our data indicates that lipid content varies among the samples by up to 1.3%.

The homogeneity of the substrate had no significant effect on glucose or total protein determination. The microorganisms can use the nutrients as shown in Table 3; the concentration of glucose was about 9.68% before anaerobic digestion, 6.68% after 5 days of retention, and about 7.19% after 10 days of retention. The difference between the last two values is not statistically significant.

At the same time, the concentration of total protein decreased during retention, being about 9.5% before anaerobic digestion and then 1.7% after biogas synthesis. This demonstrates that proteins are digested most efficiently during methanogenesis. As shown previously [31], a concentration of long-chain fatty acids in bioreactor filler that is too high can inhibit the synthesis of methane. It is also possible that glucose is being released slowly from cellulose, which is present in the CM and MA.

### 3.2. Structure of the Microbial Community during Methanogenesis

A microbial community of anaerobic sludge from a city sewage (CS) treatment plant was used for biogas production. The composition of the microbiota was determined beforehand and then again at 5 and 10 days into the anaerobic digestion process, as demonstrated in Figure 2.

Sequencing results demonstrated decreased richness and diversity of the microbial community, as well as a decreased relative abundance of bacteria in relation to archaea, similar to what has been observed previously [32]. At the same time, the presence of eukaryotic microorganisms and viruses was not observed, but another study found eukaryotes and viruses contributing 0.52% and 0.02% of abundance [33]. In sewage sludge samples (1U and 1L), the abundance of the phylum Euryarchaeota increased from 8.95% to 22.55% after the fourth retention phase. Actinobacteria, initially present in the starter mix at 10.25%, decreased to 3.4% (OLR was 2.44 kg VS/d), but rebounded to 9.6% in the final sample when OLR was 1.09 kg VS/d. The variation in Bacteroidetes was approximately 4% higher compared to samples taken before methanogenesis. The percentage of the bacteria *Candidatus cloacimonetes* varied in the samples from 4.85% initially to 6.65% and 2.80% after the fermentation process.

Percentages of another group of bacteria, *Phylum Firmicutes*, decreased from 27.35% to about 19.23% and were stable at different OLR phases (2.44 and 1.09 kg VS/d) under anaerobic conditions.

The bacteria *Phylum Synergistetes* were present at 6.35% prior to anaerobic digestion and decreased to 0.15% by the end of the process. At the order level, the abundance of Euryarchaeota OFGB 9269 increased to 6.8%, and that of the Methanomicrobiales increased to 14.4%. The abundance of Methanocorpusculaceae increased to 14.2%, while that of Methanomicrobiaceae decreased from 9.8% to 0.3%. Another study reported at the order level comparing a one-stage bioreactor with substrate was sludge from a local wastewater treatment plant in Beijing, China [33], wherein the abundance of Methanobacteriales was 1,66%, the phylum of Euryarchaeota was 2.02%, and the genus level was total. Archeal sequencing showed that when the solid anaerobic digestion batch [34] increased, Methanoculleus increased from 4.60% to 83.0%, but in our research, this methanogens decreased from 8.94% to 0%. In our research, Methanocorpusculum (taxonomy genus level) was increased from 0 to 13.65%, as noted for the species *Methanocorpusculum bavaricum.* Compared to different types of bioreactors with different types of sludge, microbiota was formed, which can produce biogas, but the main genus was not the same.

### 3.3. α and β Diversity

The evaluation of α-diversity revealed a significant difference prior to anaerobic digestion at OLRs of 2.44 and 1.09 kg VS/d. Figure 3 shows the α and β diversity of microorganism species before and after different OLRs.

The highest Shannon diversity index was observed to be 3.2 at the substrate before, and it increased by 9.6% over time and through the phases of methanogenesis when ORL was 2.44 and 1.09 kg VS/d.

Changes in the bacterial community in the three-stage bioreactor resulted in a potential rate-limiting step during anaerobic digestion. Principal Coordinate Analysis (PCoA) performed with the representative OTUs showed microbiota separation before biogas synthesis and at the different OLRs (2.44 kg VS/d and 1.09 kg VS/d) and indicated the similar composition of the bacterial microbiota (Figure 4).

Figure 4 shows that the microbiota in samples 1U and 1L before biogas production are different compared to samples taken during biogas synthesis in 2U and 2L and 3U and 3L when ORL was 2.44 kg VS/d and 1.09 kg VS/d, respectively. However, in samples 2U and 2L and 3U and 3L when the ORL was 2.44 kg VS/d and 1.09 kg VS/d, respectively the microbiota was very similar and produced methane.

### 3.4. Methane Production at the Three-Stage Bioreactor

Methanogenesis is the biological production of methane, a process of anaerobic digestion performed by a group of methanogens belonging to the Archaea domain of single-celled organisms. Because methanogenesis is the final step in the anaerobic degradation of organic carbon, the functional pathways performed by methanogens that convert acetate to CO_2_ and CH_4_ and oxidize H_2_ to H_2_O were sorted.

The correlation between methane production and the abundance of Euryarchaeota is shown in Figure 5. Cell metabolism depends on the biochemical reactions performed by various enzymes. The expression and activity of enzymes are closely related to the rate of biochemical reactions.

It was found that a 77.8% concentration of methane, when OLR was 2.44 kg VS/m^3^, correlated with a 22.65% abundance of Euryarchaeota, which was 13.65% greater than that in the starter mix. Reducing OLR to 1.09 kg VS/m^3^ decreased methane concentration to 73.8% without any significant decrease in the abundance of Euryarchaeota.

During the adaptation period, the yield of biogas and methane was 450.9 L/Kg VS_added_ and 329.6 L CH_4_/Kg VS_added_, respectively. Sufficiently high yields of biogas and methane were achieved due to the long HRT, which was 20 days. CH_4_/m^3^-d was produced between 1100 and 1374 L per day in a three-stage bioreactor at different OLRs. Although the OLR differed 2.2 times between the samples, the amount of Euryarchaeota remained similar. It was determined that when the OLR was 2.44 kg VS/d, the maximum biogas and methane yield was 223 L/kg VS_added_ and 175 L CH_4_/kg VS_added_, respectively. After the OLR was reduced to 1.09 kg VS/d, the maximum methane yield increased to 324 L CH_4_/kg VS_added_ and the biogas yield increased to 429 L/kg VS_added_. Although the concentrations of Archea Euryarchaeota at different OLRs were similar, the methane yield from 1 kg VS_added_ was higher when the OLR was 1.09 kg VS/d. This was caused by a higher VS retention time in the bioreactor, which reached 10 days.

It has been previously demonstrated [35] that 408 mL of biogas can be obtained by mixing macroalgae cultures of *Ulva rigida* with anaerobic sludge and water in a single-stage bioreactor under mesophilic conditions. During those tests, a biogas yield of 114 L/kg VS_added_ was achieved in a single-stage bioreactor, and the methane concentration reached 75%. A maximum biogas yield of 1200 L/kg VS_added_ was obtained when *Ulva rigida* was mixed with the sugar waste. In that study, the OLR reached 1.66 g VS/L-d. By mixing macroalgae with co-substrates, higher yields of biogas and methane can be obtained. A previous study [3] has shown that a maximum methane yield of 146 L/kg VS is achieved using the marine macroalgae *Saccorhiza polyschides*. The maximum concentration of methane in the biogas was 64.5% when the experiments were carried out in a single-stage bioreactor under mesophilic conditions.

In order to demonstrate the two-stage bioreactor effect [16], methane yield studies in a bioreactor using substrate from Napier grass (*Pennisetum purpureum*) were performed. During the research, it was determined that a methane yield of 282 L CH_4_/kg VS was produced, which was 30% higher than that produced by a single-stage bioreactor.

Research shows that mixing macroalgae with co-substrates such as cattle manure with straw can produce higher methane yields of up to 324 L CH_4_/kg VS_added_ in a three-stage bioreactor.

CO_2_, O_2_, and H_2_S concentrations in biogas were determined during the adaptation period and at different OLRs. CO_2_ concentrations were in the range of 20 to 30% during the research. O_2_ and H_2_S concentrations were 0–2% and 0–10 ppm, respectively.

### 3.5. Functional Pathways of Methane Production at the Three-Stage Bioreactor

Methanogenesis is the biological production of methane, a process of anaerobic respiration performed by a group of methanogens belonging to the Archaea domain of single-celled organisms. Because methanogenesis is the final step in the anaerobic degradation of organic carbon, the functional pathways performed by methanogens that convert acetate to CO_2_ and CH_4_ and oxidize H_2_ to H_2_O were sorted (Figure 6). 1200 functional pathways in total were identified; however, only 16 are related to methane biosynthesis.

The relationship between the results and the data obtained during the investigation is shown in Figure 6. Methanogens (anaerobic archaea) do not have the ability to perform a complete TCA cycle, and it is proposed that biosynthesis intermediates are synthesized through an incomplete cycle. TCA plays an important role in producing electron carriers, such as NADH and FADH2, for energy production. One of the methanogenic subsystems has a reductive tricarboxylic acid cycle (RTCA).

However, most anaerobic species use the RTCA cycle, which reduces CO_2_ and H_2_O in order to synthesize carbon compounds. Methanogenic anaerobes also have RTCA cycles. In addition, their TCA cycles are incomplete due to a lack of several steps and enzymes [36].

The abundance of pathways in different metabolic subsystems is shown in Figure 6. The dominant pathway in the city sewage sludge was the incomplete reductive TCA cycle P42-PWY. The current Macrogen database describes the P42-PWY cycle, which has seven functions provided by six enzymes (Table 4). In the order Methanomicrobiales, three families were identified: Methanomicrobiaceae, Methanocorpusculaceae, and Methanospirillaceae. All members of this order are able to produce methane from CO_2_ and H_2_. In many species, formate and secondary alcohols are used as alternative electron donors [37,38].

In this study, we identified that the most active functional pathway of methane biosynthesis is PWY-6969, which proceeds via the TCA cycle V (2-oxoglutarate synthase) (Table 4). *Methanocorpusculum bavaricum* was found to be present at 14.2% (OLR was 2.44 kg VS/d) and at 14.15% (OLR was 1.09 kg VS/d) when methane concentration and yield were highest. However, this microorganism was not found in the inoculant (city sewage).

While comparing the functional pathways between the starter culture and phases of the bioreactor (OLRs were 2.44 and 1.09 kg VS/d), it was demonstrated that methane synthesis pathways differ (Figure 6). Examples of these differing functional pathways are COA-PWY-1, the super-pathway of coenzyme A biosynthesis III (mammals); PWY-5209, methyl-coenzyme M oxidation to CO_2_; PWY-5659, GDP-mannose biosynthesis; and PWY-7851, coenzyme A biosynthesis II (eukaryotic). These pathways could be found in bacteria or in eukaryotic organisms. In the starter culture, archaea made up 9% of the total microbiota, and three species were identified. By comparison, during anaerobic digestion, when the yield of methane was highest, archaea made up 22.65% of the total microbiota, and five species were identified. At this stage, the number of functional pathways decreased by twofold compared to the starter cultures, though these pathways were more directed toward methane biosynthesis.

## 4. Conclusions

This study demonstrated that in a three-stage bioreactor, both a higher yield and concentration of methane in biogas are obtained compared to a one-stage bioreactor. The research shows that the production of biogas increases and correlates with the number of methanogens in the anaerobic reactor. Studies have shown that microbial composition changes in conjunction with changes in the OLR. It was identified that the most active functional pathway of methane biosynthesis is PWY-6969, which proceeds via the TCA cycle V (2-oxoglutarate synthase). It was found that after reducing the OLR from 2.44 to 1.09 kg VS/d, the methane yield increased from 175 L CH_4_/kg Vs_added_ to 324 L CH_4_/kg VS_added_. This change was due to the increased performance of the RT and Euryarchaeota family. By digesting macroalgae mixed with co-substrates in a three-stage bioreactor, methane-producing microorganisms can work effectively and achieve high methane yields.

## Figures and Tables

**Figure 1 microorganisms-12-00109-f001:**
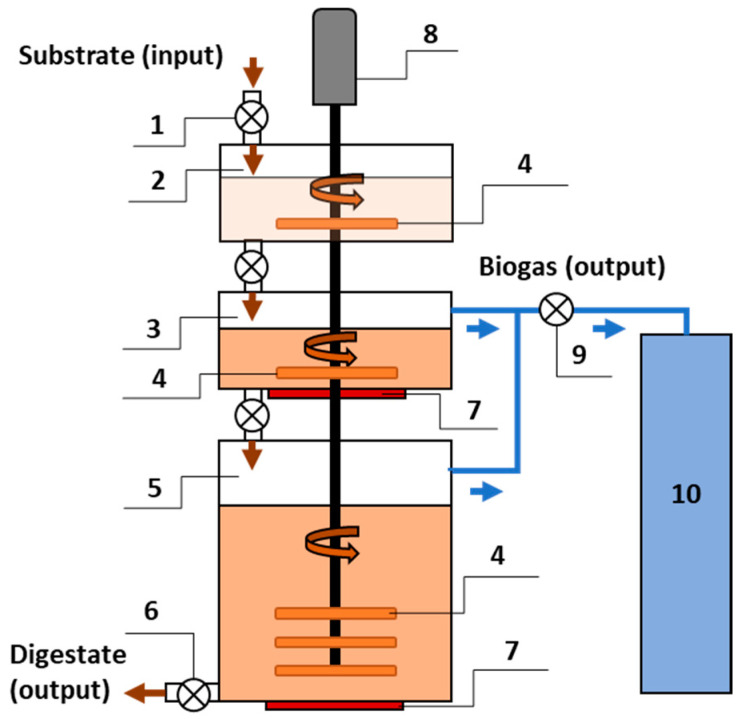
Illustration of a three-stage bioreactor: 1—substrate filling valve, 2—preparation chamber, 3—second aerobic—anaerobic chamber, 4—mixer blades, 5—third anaerobic chamber, 6—digestate outlet pipe, 7—heating elements, 8—mixer, 9—biogas valve, and 10—biogas storage tank.

**Figure 2 microorganisms-12-00109-f002:**
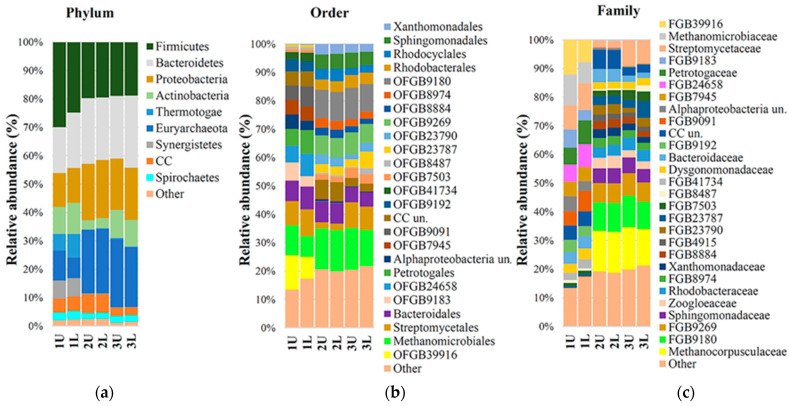
Taxonomic profiles of the microbiome from the substrate before incubation time, at 5 retention days, and at 10 retention days using 16S rRNA gene sequencing data. Current level: (**a**) Phylum; (**b**) Order; (**c**) Family. 1U and 1L were taken before incubation at the bioreactor, and 2U, 2L, 3U, and 3L were taken during methane synthesis.

**Figure 3 microorganisms-12-00109-f003:**
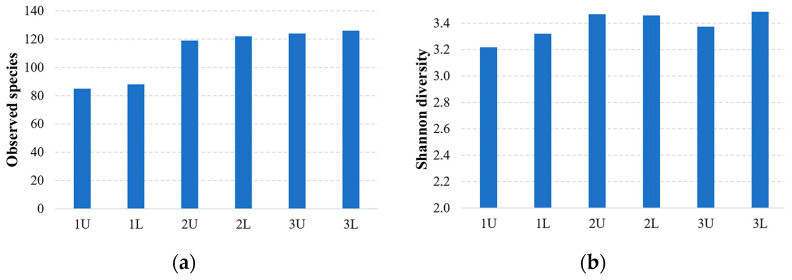
Diversity of microorganism species before stage (1U and 1L), when ORL was 2.44 kg VS/d of bioreactor (2U and 2L), and when ORL was 1.09 kg VS/d of bioreactor (3U and 3L): (**a**) alpha diversity analysis based on observed species diversity and (**b**) Shannon diversity.

**Figure 4 microorganisms-12-00109-f004:**
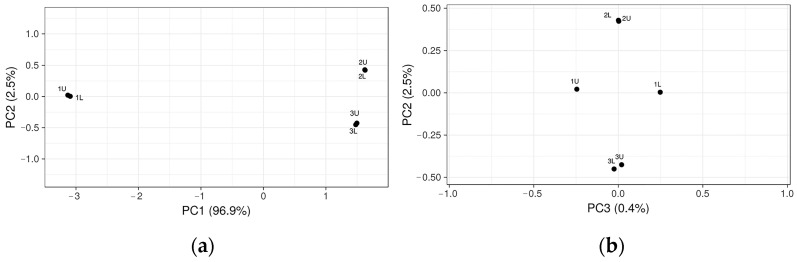
Principal Coordinate Analysis of microbiota before stage (1U and 1L), when ORL was 2.44 kg VS/d of bioreactor (2U and 2L), and when ORL was 1.09 kg VS/d of bioreactor (3U and 3L): (**a**) the similarity matrix of microbiota in coordinates PC1 and PC2 and (**b**) the similarity matrix of microbiota in coordinates PC3 and PC2.

**Figure 5 microorganisms-12-00109-f005:**
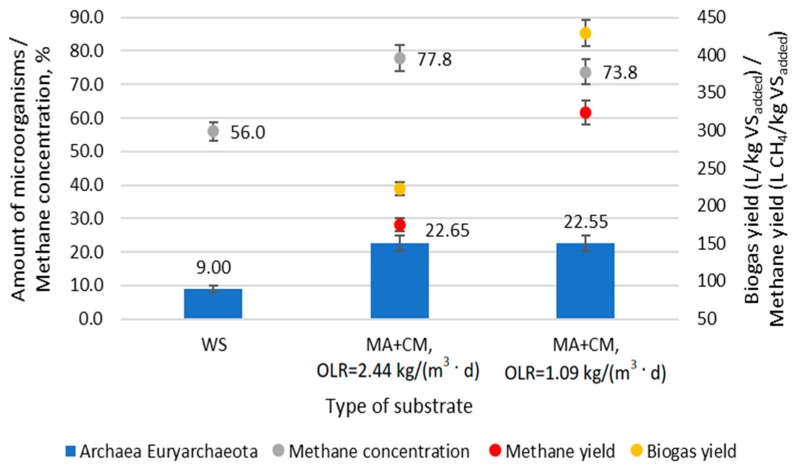
Correlation between methane yield and concentration and abundance of Euryarchaeota.

**Figure 6 microorganisms-12-00109-f006:**
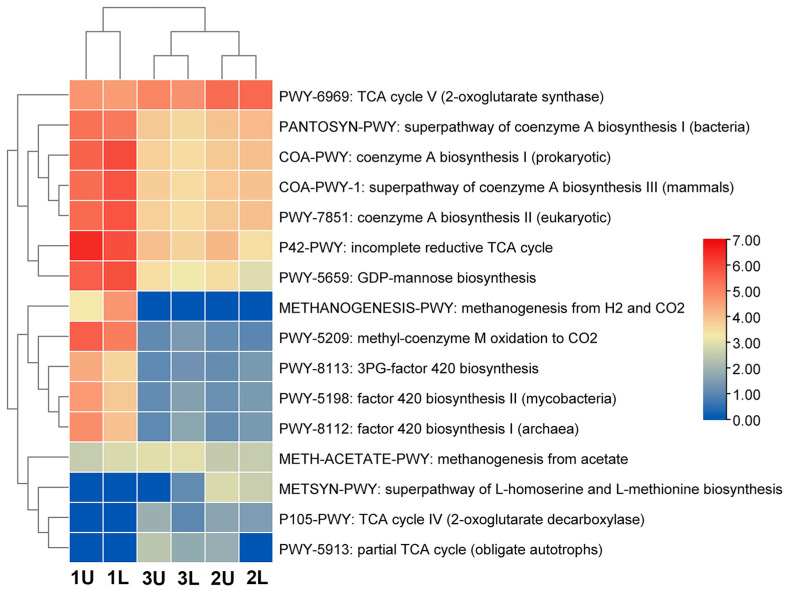
Functional pathway for methane biosynthesis in a three-stage bioreactor.

**Table 1 microorganisms-12-00109-t001:** Physicochemical parameters of substrate, co-substrate, and inoculum.

	MA (Substrate)	CM (Co-Substrate)	WS (Inoculum)
TS (g/g_wet_)	0.33 ± 0.03	0.19 ± 0.02	0.069 ± 0.007
VS (g/g TS)	0.62 ± 0.02	0.68 ± 0.02	0.035 ± 0.002
NS (g/g TS)	0.38 ± 0.01	0.32 ± 0.01	0.030 ± 0.002
C (%)	46.60 ± 0.20	52.60 ± 0.20	–
N (%)	4.44 ± 0.01	1.53 ± 0.01	–
H (%)	4.70 ± 0.01	5.30 ± 0.02	–
S (%)	0.80 ± 0.10	3.40 ± 0.20	–
C/N	10.5	34.4	–

**Table 2 microorganisms-12-00109-t002:** Three-stage bioreactor operational parameters for the AD process.

Operation Time, Days	Temperature, °C	pH	Mixing Speed, rpm	TS_added_, kg TS/d	OLR, kg VS/d	TVS, L	SC, L/d	RT, d
20	37.0 ± 0.2	7.77 ± 0.2	5.0	–	–	300.0 ± 0.1	–	20
14				4.50 ± 0.14	2.44 ± 0.07	300.0 ± 0.1	60.0 ± 0.1	5
16				2.24 ± 0.07	1.09 ± 0.04	300.0 ± 0.1	30.0 ± 0.1	10

**Table 3 microorganisms-12-00109-t003:** Concentrations of total lipids, protein, and glucose of the filler and obtained substrate before methanogenesis and at 5 and 10 days of retention.

	Total Lipids, %	Glucose, %	Total Protein, %
Before methanogenesis	1.49 ± 0.37	9.68 ± 0.89	9.47 ± 0.02
At the OLR = 2.44 kg VS/d, HRT = 5 days	2.75 ± 0.48	6.68 ± 0.6	5.19 ± 0.22
At the OLR = 1.09 kg VS/d, HRT = 10 days	1.89 ± 0.36	7.19 ± 0.75	1.68 ± 0.22

**Table 4 microorganisms-12-00109-t004:** The dominant enzymes at the P42-PWY and PWY-6969 functional pathways.

Brenda	Enzymes at to P42-PWY	Enzymes at to PWY-6969
EC 1.1.1.37	Malate dehydrogenase	Malate dehydrogenase
EC 1.1.1.42	-	Isocitrate dehydrogenase (NADP+)
EC 1.2.7.1	Pyruvate synthase	-
EC 1.2.7.3	2-oxoglutarate synthase	2-oxoglutarate synthase
EC 1.3.5.1	-	Succinate dehydrogenase
EC 2.3.3.1	-	Citrate (Si)-synthase
EC 2.3.3.9	-	Malate synthase
EC 4.1.3.1	-	Isocitrate lyase
EC 4.2.1.2	Fumarate hydratase	Fumarate hydratase
EC 6.2.1.5	Succinate-CoA ligase (ADP-forming)	Succinate-CoA ligase (ADP-forming)
EC 6.4.1.1	Pyruvate carboxylase	-

## Data Availability

Data are contained within the article.

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
