# Peer review of "Changes in Microbiota Composition during the Anaerobic Digestion of Macroalgae in a Three-Stage Bioreactor"

_microorganisms, 2024, doi:10.3390/microorganisms12010109_

Round 1
Reviewer 1 Report
Comments and Suggestions for Authors
The manuscript of Vasiliauskienė et al., investigated the microbial consortia during the anaerobic digestion of green macroalgae. The use of macroalgae is a widely studied topic that is highly relevant, especially in today’s global situation (potential climate mitigating feedstock). Therefore this paper deserves recognition in this field of research. The approach of the authors is well performed and the applied methodology merits publication in Microorganisms. However, after reading the manuscript I have some minor and major comments which ought to be addressed before publication that are mentioned below. Additionally there should be an extensive revision regarding typos, spelling and grammar mistakes. Some are mentioned below, but not all of them. The reviewer suggests a proofread by a native English speaker.
- - To not overcomplicate things, the reviewer suggests to stick with macroalgae in the introduction section. The manuscript involves the valorization of macroalgae, however, the introduction section starts with the microalgae Chlorella.
- - Rephrase statement in Line 51. The reviewer assumes that the authors mean “Macroalgae as a feedstock for biogas are a good alternative to fossil fuels”? As Macroalgae themselves are not a direct substitute for fossil fuels.
- - The authors mention the benefits and advantages of the use of macroalgae, specifically for anaerobic digestion in this case. However, there are also clear disadvantages (or challenges) associated with the use of macroalgae in anaerobic digestion. For instance salinity can be a serious hurdle or the highly seasonal variance to name a few. Please also mention this in the introduction section.
- - Typo Line 118.
- - Line 135: Microalgae or Macroalgae?
- - Typo Line 137.
- - Line 138: Microalgae or Macroalgae?
- - Line 144: Microalgae or Macroalgae?
- - Line 145: place numbers in subscript.
- - What was the pre-processing of the used green seaweed feedstock? The seaweed was harvest and immediately used? Or was it first (partially) dried? Frozen? Kept in the refrigerator upon usage? Please elaborate.
- - Was the seaweed biomass as such administered to the first reactor? Or was is already partially chopped into larger chunks? Please elaborate.
- - The aim of the first reactor is to mix and homogenize the substrate. What kind of particle sizes are fed into the second reactor? What are the criteria here?
- - Please give the details of the chromatographic method that is used to determine the concentration of the different gases. Which gas is measured with GFM 410 and which one with GC? What is the detector of the GC? Please elaborate.
- - As there is a lot of nitrogen present in the initial feedstock, did the authors check for ammonia/ammonium production?
- - Typo Line 255.
- - Please mind significant figures throughout the text.
- - Typo Line 271.
- - Typo Line 272.
- - Typo Line 273.
- - Typo Line 278.
- - Typo Line 303.
- - Line 309, “3” should be in superscript.
- - Please remove grey line (box) around Figure 5 and place “4” in subscript in y secondary axis. Place 3 in superscript in x axis.
- - Typo Line 317.
- - Typo Line 320.
- - Typo Line 323.
- - Typo Line 333.
- - Typo Line 343.
- - A graph concerning the cumulative biogas production overtime would be appreciated.
- - The authors mention in the analytical section that CH4, CO2, H2S … are measured. However in the discussion of the results the values of these measurements (besides CH4) are nowhere mentioned.
Comments on the Quality of English LanguageSee comments above.
Author Response
Thank you for your relevant comments, we are sending comments. The corrections are shown in blue in the text. Please see the attachment.

Reviewer 2 Report
Comments and Suggestions for Authors
In general, the article does not contain any serious flaws; both methods used and results obtained correspond to goals and conclusions, respectively.
The following minor edits are required:
Line 26 – VS. Decipher the abbreviation in the abstract.
Line 111 – DS. Check the abbreviation.
Line 125 – denoted as 1U and 1L, and subsequently at third (samples 2U and 2L) and fourth (samples 3U 3L) retention phases. Check the phases and designations.
Line 152 – Only one electric motor is used for biomass mixing, i.e. biomass in all chambers is mixed simultaneously. Add explanation considering mixing (continuous mode?).
In general, experiment description should be improved.
1. The scheme of the reactor is not obvious. Could you add photo and/or diagram, which demonstrated chambers, inlets/outlets, etc?
2. Did you start the process in a batch mode?
3. Could you include the data demonstration adaptation of the process to continuous mode?
4. Did you control the product parameters after each chamber (or only after the third stage)?
5. Did you collect the samples for metagenomic analysis in the third chamber?
6. 2.2. Measurement of the protein, total lipids and glucose concentration. Line 126: We conducted an analysis of a substrate at various stages, focusing on the upper (U) and lower (L) layers.
Could you explain designations “upper” and “lower” layers? Were both layers in the chamber 3?
7. Could you add summarizing diagram of your experiments to clarify forementioned points and designate retention phases?
Author Response

(The authors gave the same response as above.)

Reviewer 3 Report
Comments and Suggestions for Authors
This paper, entitled Changes in Microbiota Composition During the Anaerobic Digestion of the Macroalgae in a Three-Stage Bioreactor, is a scholarly work and can increase knowledge in this domain. The authors provide an interesting and original study, the content is relevant to Microorganisms.
I have some general and specific comments:
- The abstract and keywords are meaningful.
- The manuscript is wuite well written and well related to existing literature.
- Pleack check significant number of digits of accuracy for data in tables.
- Figure 1 is not essential and could be removed, this figure should be rearranger, as it this figure is not very informative and could be considered as Graphical abstract instead of figure in the main text.
- Please provide accuracy of data in Table 2.
- Considering three stage is quite confusing due to the fact that the first stage is for biomass preparation, it's not a fermentative step as usual considered in AD processes.
- Please provide dimensions, size, operational parameters for AD process.
- Please discuss about microbiota composition and change during this study and with other existing studies dealing with AD process of other substrates. There's some studies in literature dealing with similar consideration. Please compare and discuss the data and results.
As it, this paper is not fully acceptable for publication and requires major amendment and additional data or information. I recommend the following decision: RECONSIDER AFTER MAJOR REVISION.
Author Response

(The authors gave the same response as above.)

Round 2
Reviewer 1 Report
Comments and Suggestions for Authors
The authors addressed most of the comments raised by the reviewer and therefore, the reviewer accepts this manuscript for publication in Microorganisms.
Reviewer 2 Report
Comments and Suggestions for Authors
Authots have improved the manuscript according to the review. Therefore, it may be accepted.
Reviewer 3 Report
Comments and Suggestions for Authors
The authors provide a revised version of their manuscript taking into account all the comments and request of amendments made in the previous review. The authors provide also detailed and justified answers for all comments. I agree with all the answers. I recommend the following decision: ACCEPT IN PRESENT FORM.